# The effect of myalgic encephalomyelitis/ chronic fatigue syndrome (ME/CFS) severity on cellular bioenergetic function

**Cara Tomas** [1]\*, **Joanna L. Elson**[1,2], **Victoria Strassheim**[3], **Julia L. Newton**[1,3], **Mark Walker**[1]

**1** Translational and Clinical Research Institute, Newcastle University, Newcastle upon Tyne, United Kingdom, **2** Centre for Human Metabolomics, North-West University, Potchefstroom, South Africa, **3** Newcastle upon Tyne Hospitals, NHS Foundation Trust, Newcastle upon Tyne, United Kingdom

\* cara.tomassmith@nhs.net

**Data Availability Statement:** All relevant data are within the manuscript and its Supporting Information files.

## Abstract

Myalgic encephalomyelitis/ Chronic fatigue syndrome (ME/CFS) has been associated with abnormalities in mitochondrial function. In this study we have analysed previous bioenergetics data in peripheral blood mononuclear cells (PBMCs) using new techniques in order to further elucidate differences between ME/CFS and healthy control cohorts. We stratified our ME/CFS cohort into two individual cohorts representing moderately and severely affected patients in order to determine if disease severity is associated with bioenergetic function in PBMCs. Both ME/CFS cohorts showed reduced mitochondrial function when compared to a healthy control cohort. This shows that disease severity does not correlate with mitochondrial function and even those with a moderate form of the disease show evidence of mitochondrial dysfunction. Equations devised by another research group have enabled us to calculate ATP-linked respiration rates and glycolytic parameters. Parameters of glycolytic function were calculated by taking into account respiratory acidification. This revealed severely affected ME/CFS patients to have higher rates of respiratory acidification and showed the importance of accounting for respiratory acidification when calculating parameters of glycolytic function. Analysis of previously published glycolysis data, after taking into account respiratory acidification, showed severely affected patients have reduced glycolysis compared to moderately affected patients and healthy controls. Rates of ATP-linked respiration were also calculated and shown to be lower in both ME/CFS cohorts. This study shows that severely affected patients have mitochondrial and glycolytic impairments, which sets them apart from moderately affected patients who only have mitochondrial impairment. This may explain why these patients present with a more severe phenotype.

## Introduction

Myalgic encephalomyelitis/chronic fatigue syndrome (ME/CFS) is a highly heterogeneous disease with an unknown aetiopathogenesis. There are around 250,000 patients in the UK alone

**Funding:** This study was funded by an ME Research UK grant to MW. Additional funding awarded to JLN from The Medical Research Council, Action for ME, and the ME Association was also used to fund this study.

**Competing interests:** The authors have declared that no competing interests exist.

with an estimated 25% of patients being severely affected by the disease [1, 2]. Symptoms include, but are not limited to, severe fatigue lasting more than 6 months, post-exertional malaise, memory and concentration problems and disrupted sleep [3]. ME/CFS has a marked impact on the quality of life of patients [4, 5]. Our group has previously used extracellular flux analysis to show that PBMCs from ME/CFS patients have significantly lower mitochondrial function than healthy controls [6]. Bioenergetic profiling using blood samples is not a new technique and has been utilised in other studies looking at a range of disease types [7–10]. The techniques required for investigating mitochondrial function in blood cells are minimally invasive but provides a wealth of data than allows further understanding of the pathophysiology of disease [11]. Blood cells are found systemically therefore it is thought that the bioenergetic profile of these cells is representative of the body as a whole [12]. However, it is yet to be determine whether mitochondrial changes in the blood are a cause or consequence of disease. The ME/CFS cohort in our previous study was comprised of moderately and severely affected ME/CFS patients but critically did not look to see if the decline in bioenergetic function was correlated with disease severity. In this study we re-examined the previously published data in order to investigate the role of mitochondrial and glycolytic function in peripheral blood mononuclear cells (PBMCs) in disease severity in ME/CFS. Critically there are very few studies in the field of ME/CFS research include severely affected patients. This is due to the patients in this group being deemed 'hard to reach' as they are housebound or even bedbound by the disease. The inability of clinicians and researchers to access these patients means that they are excluded from the majority of ME/CFS studies despite their pressing clinical need. This manuscript is the first study to look at cellular bioenergetics in severe ME/CFS patients and investigate if they differ from those moderately affected with the disease.

This study also looks at the relative contribution of glycolytic acidification to overall extracellular acidification and how it impacts on the calculation of glycolytic parameters. The glycolysis stress test by Agilent Technologies, used to measure extracellular acidification rate (ECAR) and calculate the glycolytic rates of cells, is based on the presumption that the majority of protons being moved into the extracellular medium are produced from the conversion of pyruvate to lactate and the subsequent extrusion of protons from the cell. However, in some cell types this may not be the case. It has previously been shown that the percentage contribution of respiratory acidification can vary vastly between cell types with results showing a percentage contribution of between 0–100% for different cell types [13]. Respiratory acidification is caused by the production of $CO_2$ from substrate oxidation. The $CO_2$ is hydrated to carbonic acid which subsequently dissociates to form bicarbonate and a proton. In the glycolysis stress test, respiratory acidification can be wrongly attributed to cellular glycolysis as the glycolysis stress test cannot discriminate between glycolytic acidification and respiratory acidification when used as directed by the manufacturers. The use of an equation described by Mookerjee et al allows the percentage contribution of respiratory acidification and glycolytic acidification to be taken into account when calculating glycolytic parameters from a glycolysis stress test [13].

Additionally, we performed a calculation of the ATP-linked respiration rate of PBMCs in the ME/CFS and healthy control cohorts, devised by Mookerjee et al, which enabled us to determine the individual contribution of glycolysis and oxidative phosphorylation (OXPHOS) to cellular ATP-linked respiration rates [14]. We were then able to investigate how extracellular glucose concentrations affected the balance between glycolytic and OXPHOS derived ATP.

This work describes novel investigations into the functioning of energy production pathways in ME/CFS cells and is unique in its inclusion of a severely affected cohort in bioenergetics studies for the first time.

## Materials and methods

### Participant information

ME/CFS patients were diagnosed by a single physician (JLN) using the Fukuda criteria [3], but all patients also adhere to the Canadian Consensus Criteria [15]. Ethical approval for the sample collection from ME/CFS patients was granted from the National Research Ethics Committee North East–Newcastle & North Tyneside 2. Blood samples from healthy controls were collected through the Institute of Cellular Medicine (Newcastle University) blood study with ethical approval from the National Research Ethics Committee North East–County Durham & Tees Valley. All samples were gathered after informed written consent was obtained.

There were two cohorts of ME/CFS patients with differing levels of disease severity included in this study. The moderately affected cohort had reduced mobility due to the disease but were able to attend the local fatigue clinic. Most of the moderately affected patients were able to walk into the clinic unaided but a small proportion needed the assistance of crutches or a wheelchair. The severely affected cohort were either housebound or bedbound by the disease and were unable to attend the clinic even with the use of a wheelchair. This is in line with the recognised classifications of disease severity in ME/CFS as outlined in the NICE guidelines and the International Consensus Criteria (ICC) [16, 17]. ME/CFS symptom severity is defined by the reduction in activity compared to pre-illness. Moderately affected patients are mostly housebound by the disease while severely affected patients are mostly bedridden [17]. These seemingly arbitrary categories exist due to the absence of biomarkers in the disease that would be able to distinguish between the disease severity groups and allow more robust definitions to be applied.

As severely affected patients were housebound or bedbound by the disease, blood sample collection happened at the homes of the patients by a nurse. This is in contrast to the moderately affected patients and the healthy controls who had blood collected in a clinical setting. When setting up the study to collect blood samples from severely affected patients we were aware of how fatiguing the patients could find such a visit and therefore decided, in order to minimise the effect on patients, that no other patient information would be collected at the time of the visit. Therefore we are unable to compare patient demographics between the groups used here.

### PBMC preparation

Blood samples were prepared and PBMCs isolated as described previously in Tomas et al. [6]. Briefly, a density gradient was created using Histopaque® density gradients and the PBMC layer collected. PBMCs were then washed twice in fresh PBS and resuspended in RPMI-1640 medium supplemented with 10% FBS and 1% penicillin-streptomycin. Cells were then frozen at -80˚C in 50% supplemented RPMI-1640 and 50% freezing medium (40% FBS, 10% DMSO). For cell revival, vials were rapidly defrosted in a water bath at 37˚C and added to 10ml of fresh supplemented RPMI-1640. Cells were centrifuged at 700 x g for 10 minutes and cells resuspended in supplemented RPMI-1640. Trypan blue was used to determine cell viability. All blood samples were processed within 4 hours of blood collection.

### Extracellular flux analysis

Extracellular flux analysis was conducted previously as described in Tomas et al. [6], and the data re-analysed with different sub-groups.

For mitochondrial respiration the oxygen consumption rate (OCR) of cells was measured using the Seahorse XF$^e$96 extracellular flux analyser as per the manufacturer's instructions

[18]. This included sequential additions of oligomycin, FCCP, and rotenone and antimycin A as described in Tomas et al. The following parameters were calculated: basal respiration, ATP-linked respiration, proton leak, maximal respiration, reserve capacity, non-mitochondrial respiration, and coupling efficiency. Basal respiration is; basal OCR–rotenone and antimycin A OCR. ATP-linked respiration is; basal OCR–oligomycin OCR. Proton leak is; oligomycin OCR–rotenone and antimycin A OCR. Maximal respiration is; FCCP OCR–rotenone and antimycin A OCR. Reserve capacity is; FCCP OCR–basal OCR. Non-mitochondrial respiration is; rotenone and antimycin A OCR. Coupling efficiency is; (ATP-linked respiration/basal respiration)*100.

For glycolytic parameters the extracellular acidification rate (ECAR) of cells was measured using the Seahorse XF$^e$96 extracellular flux analyser as per the manufacturer's instructions [19]. Sequential additions of glucose, oligomycin, and 2-deoxy glucose (2DG) were made as per manufacturer's instructions. From this experiments glycolysis, glycolytic capacity, glycolytic reserve, and non-glycolytic acidification were measured. The parameters were calculated as follows: glycolysis; glucose ECAR –2-DG ECAR. Glycolytic capacity; oligomycin ECAR–2-DG ECAR. Glycolytic reserve; glycolytic capacity–glycolysis. Non-glycolytic acidification; 2-DG ECAR.

All data was normalised to protein content per well using a bicinchoninic acid (BCA) assay.

Data showing mitochondrial and glycolytic function of PBMCs was examined in order to take into account disease severity.

## Extracellular acidification analysis

Extracellular acidification is attributable to either respiratory or glycolytic activity. These two parameters add up to the total proton production rate (PPR$_{tot}$) measured in cells. PPR is ECAR (the unit of measurement on the seahorse extracellular flux analyser) divided by the buffering power (BP) of the media used in the experiment (which is 0.1) [20].

$$PPR_{tot} = ECAR_{tot}/BP$$

In this study we have taken into account glycolytic acidification when calculating cellular glycolytic parameters. Glycolytic acidification was calculated using the following equation published my Mookerjee et al [13]:

$$PPR_{glyc} = ECAR_{tot}/BP - (10^{(pH-pK1)}/(1 + 10^{(pH-pK1)}))(max\ H^+/O_2)(OCR_{tot} - OCR_{rot/myx})$$

Respiratory acidification is the total proton production rate (PPR$_{tot}$) measured on the extracellular flux analyser minus the PPR$_{glyc}$.

$$PPR_{resp} = PPR_{tot} - PPR_{glyc}$$

This study also looks at the ATP-linked respiration rates of cells using the samples from the previously published work. Using equations described in [14] we calculated total ATP-linked respiration rate, glycolysis ATP-linked respiration rate, OXPHOS ATP-linked respiration rate, and % contribution of glycolysis to overall ATP-linked respiration rate of PBMCs from ME/CFS patients and healthy controls. The equations for calculating glycolysis ATP-linked respiration rate (ATP$_{glyc}$) and OXPHOS ATP-linked respiration rate (ATP$_{OX}$) are shown below:

$$ATP_{glyc} = (PPR_{glyc} * ATP/lactate) + (OCR_{mito} * 2P/O_{glyc})$$

$$ATP_{OX} = (OCR_{coupled} * 2P/O_{oxphos}) + (OCR_{mito} * 2P/O_{TCA})$$

For more information about these equation please see reference [14]. Total ATP-linked respiration rate ($ATP_{tot}$) is:

$$ATP_{tot} = ATP_{glyc} + ATP_{OX}$$

## Statistics

Where two cohorts were compared, student's or Welch's t-tests were used to compare groups depending on whether variances were equal or not. Post hoc Bonferroni corrections were used when required. Where three cohorts were compared, one-way ANOVAs were used with post-hoc Tukeys. For the comparison of three cohorts with two different glucose conditions two-way ANOVAs were applied. Correlations between multiple pairs of bioenergetic parameters from the mitochondrial stress test were calculated using multi-variate analysis in SPSS. Significance levels were set for r-values ($r > 0.4$) and p-values ($p < 0.01$).

## Results

### Mitochondrial function is impaired in ME/CFS regardless of disease severity

The cellular respiration data shown in Tomas *et al* was re-analysed to determine if disease severity affected mitochondrial function [6]. Using the previously published data we separated the ME/CFS cohort into moderately and severely affected patient cohorts and compared mitochondrial parameters from the two groups and also compared the two patient cohorts to the healthy control cohort previously published in Tomas *et al* [6]. Results are shown in Fig 1.

For the cellular respiratory parameters, one-way ANOVA showed there to be differences between groups for the parameters of basal respiration, ATP-linked respiration, proton leak, maximal respiration, and reserve capacity ($p < 0.001$). When groups were further analysed using post-hoc Tukeys test and Bonferroni correction, it was shown that both the moderately and severely affected groups were significantly lower in terms of all five parameters than the healthy controls ($p \leq 0.05$). This is in keeping with the previously published data which showed the ME/CFS cohort as a whole to have significantly lower values for these five parameters [6].

Parameters measured in this assay are inter-dependent with some parameters calculated from others [18, 21]. To see if the relationships between the parameters differ between the cohorts used in this study we performed multivariate analysis. Results from this analysis are shown in Fig 2 and Table 1. As used by other groups in previous publications, we applied minimum threshold values of 0.4 for the r-value and <0.01 for the p-value [22].

The correlation between basal respiration and ATP-linked respiration are very strong for all three cohorts in the study. This shows the tight coupling between these two parameters is maintained even in the severely affected patient group. Some of the parameters from the mitochondrial stress test are calculated using other parameters so relationships between some parameters may be expected, for example basal respiration is comprised of ATP-linked respiration and proton leak. This is reflected in the results shown in Fig 2 and Table 1. Graphs for all other correlations can be seen in S1 Appendix. Together the data from the parameter correlations shows that the relationships between some of the parameters differs between the three cohorts with the only parameters consistently correlated in all three cohorts being basal respiration and ATP-linked respiration. Further experimental work is needed to investigate why and how certain relationships become disrupted in ME/CFS PBMCs.

### Glycolytic acidification is lower in severely affected ME/CFS patients

The percentage contribution of glycolytic acidification of PBMCs was calculated for the healthy control cohort, the ME/CFS cohort as a whole, and the moderately and severely affected patient cohorts. Results are shown in Fig 3.

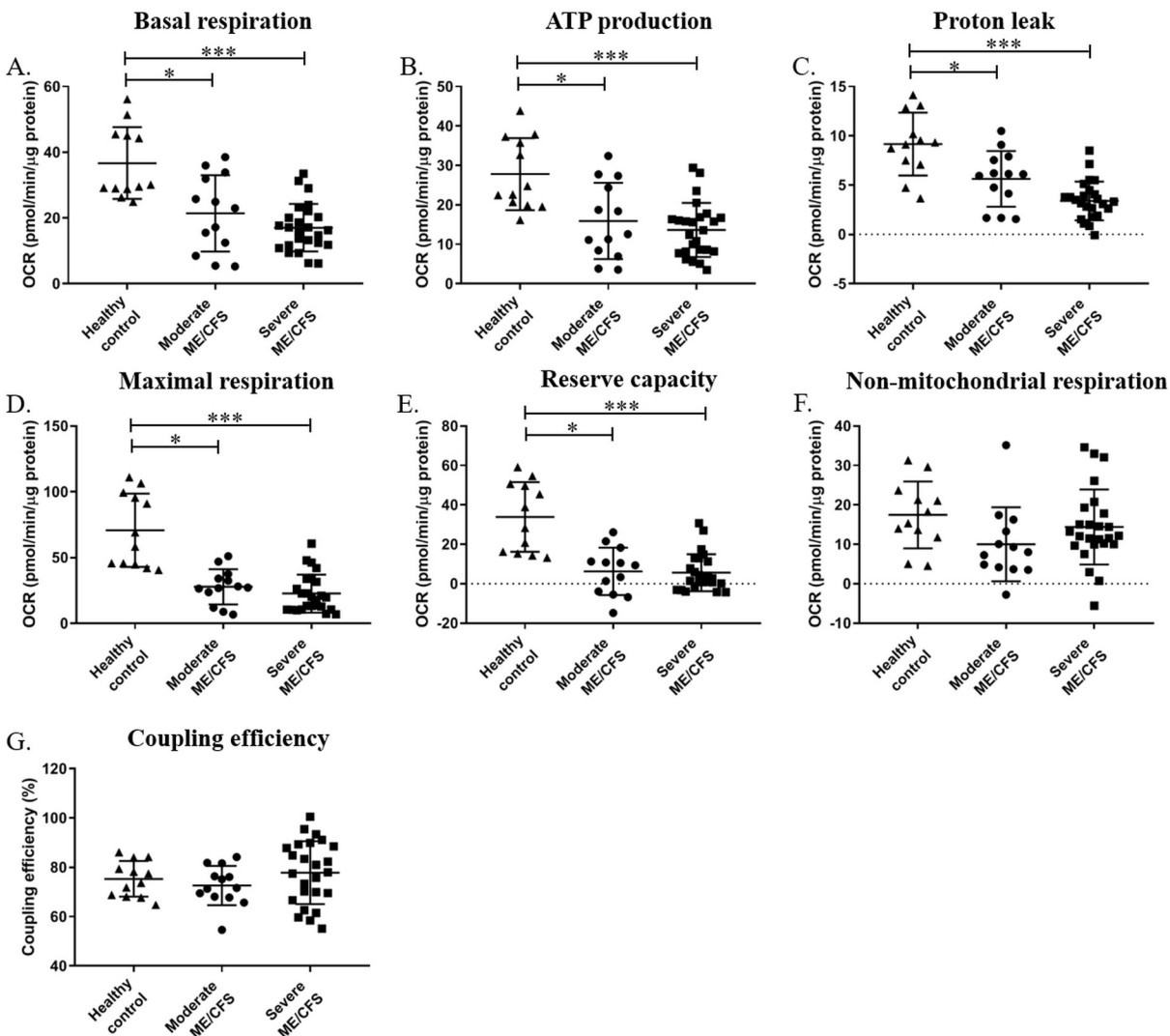

**Fig 1. Parameters of cellular respiration (A-G) and glycolytic function (H-K) in PBMCs from moderately and severely affected ME/CFS patients, and healthy controls.** (**A**) Basal respiration. (**B**) ATP-linked respiration. (**C**) Proton leak. (**D**) Maximal respiration. (**E**) Reserve capacity. (**F**) Non-mitochondrial respiration. (**G**) Coupling efficiency. Groups were compared using one-way ANOVAs followed by post-hoc Tukeys and Bonferroni correction. Healthy controls n = 12; moderately affected ME/CFS n = 13; severely affected ME/CFS n = 25. $^*p \leq 0.05$; $^{***}p \leq 0.001$.

Glycolytic acidification of PBMCs from the ME/CFS cohort as a whole was compared to that of the healthy controls (Fig 3A). Significant differences in the percentage contribution of glycolytic acidification to overall extracellular acidification were seen between the two groups with ME/CFS patients having a lower contribution of glycolytic acidification compared to healthy controls (p<0.001). However, when the ME/CFS cohort is separated out into the moderately and severely affected groups it becomes clear that the results from the ME/CFS cohort as a whole are being skewed by the results from the severely affected cohort (Fig 3B). There is a decrease in the percentage of overall extracellular acidification that can be attributed to glycolytic acidification in the severely affected CFS PBMCs when compared to both the healthy control and moderately affected ME/CFS cohorts (p≤0.002). The decrease in contribution of glycolytic acidification to overall extracellular acidification in the severely affected group is due

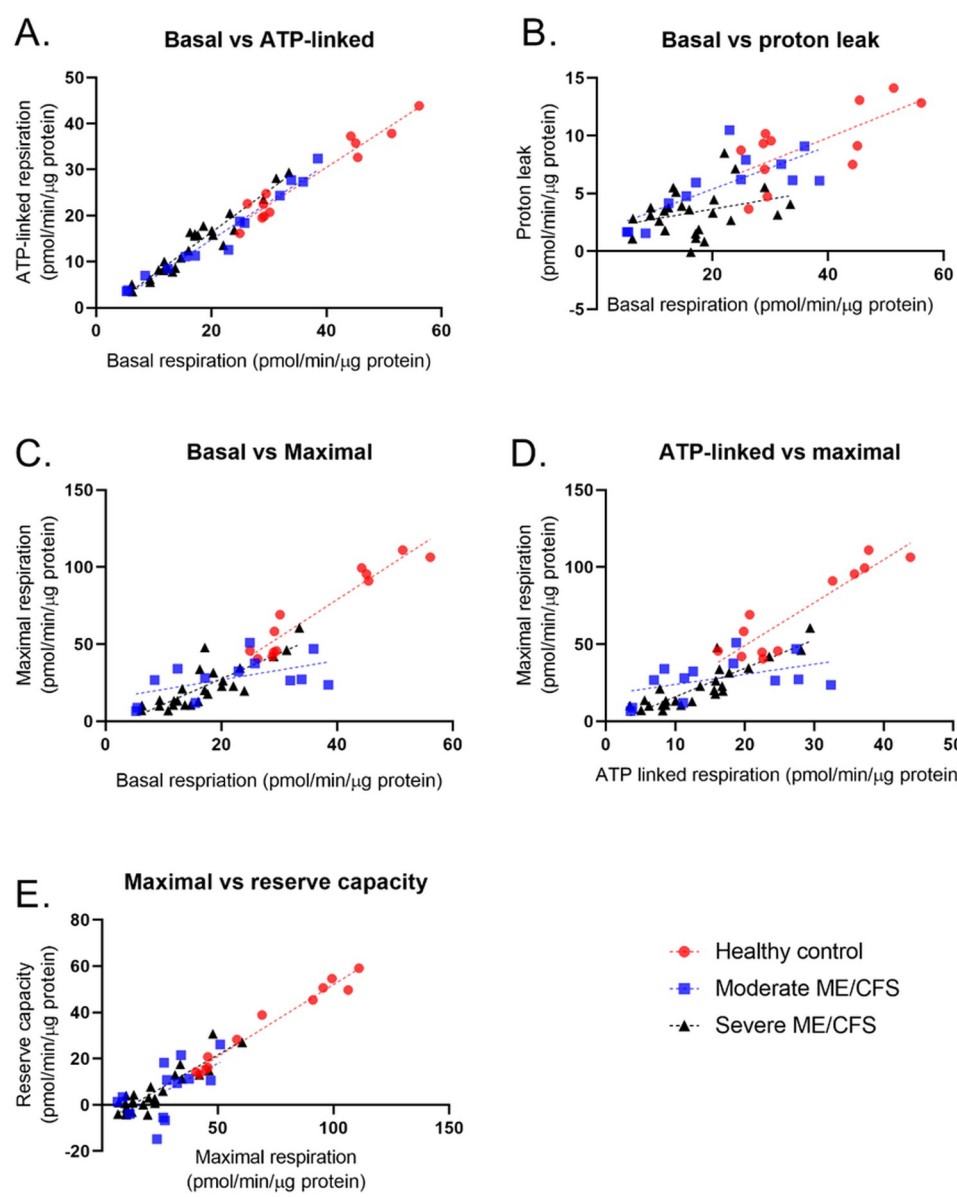

**Fig 2. Correlations between mitochondrial stress test parameters in PBMCs from healthy controls (n = 12), moderately affected ME/CFS patients (n = 13), and severely affected ME/CFS patients (n = 25).** (A) Correlation between basal respiration and ATP-linked respiration. (B) Correlation between Basal respiration and proton leak. (C) Correlation between basal respiration and maximal respiration. (D) Correlation between ATP-linked respiration and maximal respiration. (E) Correlation between maximal respiration and reserve capacity.

to a larger percentage of respiratory acidification. The moderately affected and healthy controls cohorts have comparable levels of glycolytic acidification (p = 0.257) (Fig 3B). It should be noted that it is a sub-group of severely affected patients that show a decreased glycolytic acidification with many of the severely affected cohort showing comparable levels to the moderately effected cohort. $CO_2$ acidification, or respiratory acidification, contributes substantially to the total cellular acidification in severely affected PBMCs and therefore should be taken into account when measuring ECAR in PBMCs. This shows the importance of separating respiratory and glycolytic acidification using the equations devised by Mookerjee et al [13].

**Table 1. R-values and correlation p-values for mitochondrial stress test parameters comparisons in PBMCs from healthy controls, moderately affected ME/CFS patients, and severely affected ME/CFS patients.**

**Correlation r-values:**

| | Basal respiration | | | ATP-linked respiration | | | Proton leak | | | Maximal respiration | | | Reserve capacity | | |
|---|---|---|---|---|---|---|---|---|---|---|---|---|---|---|---|
| | HC | Mod | Sev | HC | Mod | Sev | HC | Mod | Sev | HC | Mod | Sev | HC | Mod | Sev |
| Basal | | | | | | | | | | | | | | | |
| AL | **0.940** | **0.967** | **0.926** | | | | | | | | | | | | |
| PL | **0.475** | **0.583** | 0.099 | 0.242 | **0.401** | 0.002 | | | | | | | | | |
| Maximal | **0.911** | 0.295 | **0.684** | **0.831** | 0.225 | **0.778** | **0.491** | **0.400** | 0.002 | | | | | | |
| RC | **0.780** | 0.134 | 0.252 | **0.697** | 0.182 | 0.379 | **0.452** | 0.002 | 0.092 | **0.966** | 0.339 | **0.812** | | | |
| NM | 0.114 | 0.182 | **0.286** | 0.168 | 0.208 | 0.269 | 0.002 | 0.033 | 0.024 | 0.167 | 0.064 | **0.502** | 0.195 | 0.017 | 0.460 |

**Correlation probability:**

| | Basal respiration | | | ATP-linked respiration | | | Proton leak | | | Maximal respiration | | | Reserve capacity | | |
|---|---|---|---|---|---|---|---|---|---|---|---|---|---|---|---|
| | HC | Mod | Sev | HC | Mod | Sev | HC | Mod | Sev | HC | Mod | Sev | HC | Mod | Sev |
| Basal | | | | | | | | | | | | | | | |
| AL | **<0.0001** | **<0.0001** | **<0.0001** | | | | | | | | | | | | |
| PL | 0.013 | **0.002** | 0.125 | 0.104 | 0.020 | 0.833 | | | | | | | | | |
| Maximal | **<0.0001** | 0.055 | **<0.0001** | **<0.0001** | 0.102 | **<0.0001** | 0.011 | 0.020 | 0.852 | | | | | | |
| RC | **<0.0001** | 0.218 | **0.011** | **0.001** | 0.147 | **0.001** | 0.017 | 0.892 | 0.140 | **<0.0001** | 0.037 | **<0.0001** | | | |
| NM | 0.283 | 0.146 | **0.006** | 0.185 | 0.117 | **0.008** | 0.901 | 0.554 | 0.464 | 0.187 | 0.406 | **<0.0001** | 0.151 | 0.676 | **<0.0001** |

Parameters in bold meet the threshold values of 0.4 for r-values and 0.01 for p-values. AL (ATP-linked); PL (proton leak); RC (reserve capacity); NM (non-mitochondrial respiration); HC (healthy control); Mod (moderately affected ME/CFS); Sev (severely affected ME/CFS).

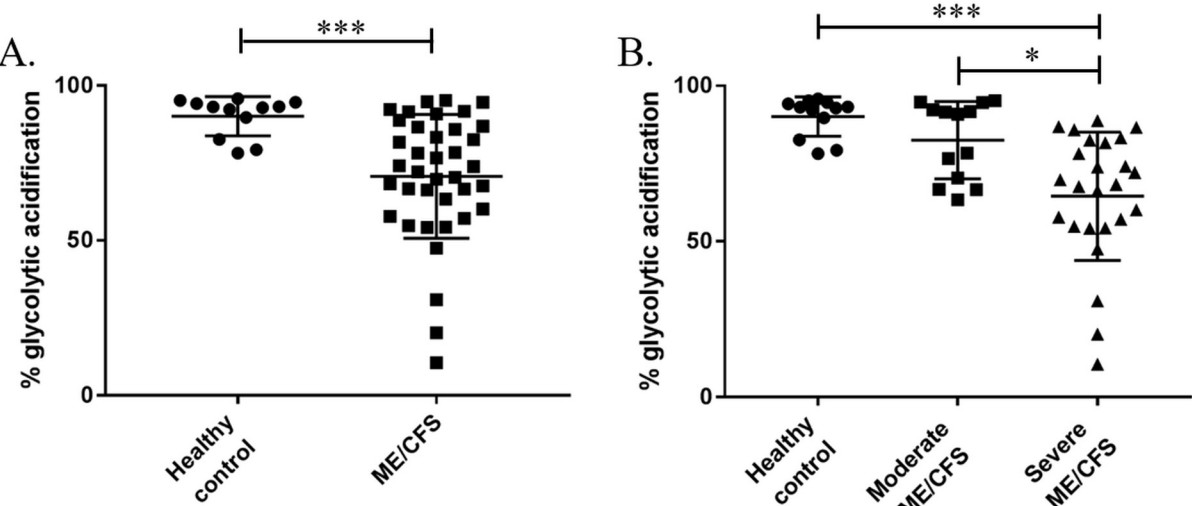

**Fig 3. The percentage of overall extracellular acidification that can be attributed to glycolytic acidification in control and ME/CFS PBMCs. (A)** Comparing percentage contribution of glycolytic acidification in PBMCs between a healthy control cohort and an ME/CFS cohort (containing both moderately and severely affected patients). **(B)** Comparison of percentage contribution of glycolytic acidification in PBMCs from a healthy control cohort, moderately affected ME/CFS cohort, and a severely affected ME/CFS cohort. Groups in (A) were compared using Welch's t-tests with post-hoc Bonferroni correction. Groups in (B) were compared using a one-way ANOVA with post-hoc Bonferroni correction. Healthy controls n = 12; whole ME/CFS cohort n = 38; moderately affected ME/CFS n = 13; severely affected ME/CFS n = 25. $^{*}p \leq 0.01$; $^{***}p \leq 0.001$.

## Glycolysis is lower in severely affected ME/CFS patients

Our group has previously published data using PBMCs from ME/CFS and healthy control cohorts to compare glycolytic function of cells using extracellular flux analysis [6]. Glycolysis was calculated by measuring the extracellular acidification rate after the addition of glucose to the cells as glycolysis produces protons which are transported to the extracellular medium. Our previous publication showed ME/CFS and healthy control PBMCs to have comparable rates of glycolysis. Fig 3 showed there to be higher levels of respiratory acidification in a sub-group of the severely affected ME/CFS cohort meaning that the original results may not accurately reflect glycolytic activity within the cells as it assumed a minimal contribution of respiratory acidification. Re-analysis of the previously published data, in order to take into account the true contribution of glycolytic acidification to overall ECAR, showed there to be no significant differences between in the PBMCs from the ME/CFS cohort as a whole and the healthy control cohort for any of the glycolytic parameters calculated from a glycolysis stress test (p≥0.263) (S2 Appendix).

There were no significant differences in glycolysis between any of the groups when the ME/CFS cohort was separated into moderately and severely affected cohorts (Fig 4A). This is consistent with the previous results from our group which found no significant differences in glycolysis when all of the ME/CFS patients were analysed as a single cohort [6]. However, when the moderately and severely affected ME/CFS cohorts were separated and respiratory acidification taken into account, the severely affected cohort were shown to have significantly lower levels of glycolysis than the moderately affected and healthy control cohorts (p≤0.036) (Fig 4B). These significant differences between cohorts is caused by the lower levels of glycolytic acidification seen in the severely affected group (Fig 3B). When the ME/CFS cohort was separated into moderately and severely affected cohorts and assessed before and after adjusting for respiratory acidification, there were no differences seen between any of the groups for the other three glycolysis parameters—glycolytic capacity, glycolytic reserve, and non-glycolytic acidification (supplementary data).

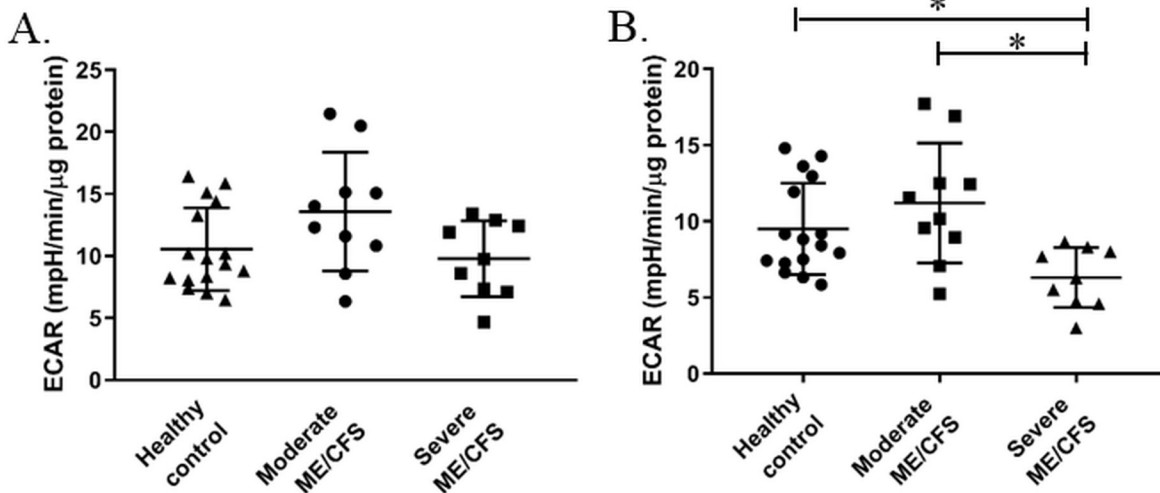

**Fig 4. Glycolysis in PBMCs from healthy controls and moderate and severely affected ME/CFS patients (published in [6]) before and after adjusting for the contribution of respiratory acidification to overall extracellular acidification. (A)** Glycolysis in PBMCs from healthy controls (n = 16), moderately affected ME/CFS patients (n = 10), and severely affected ME/CFS patients (n = 9). **(B)** Glycolysis adjusted for respiratory acidification. Groups were compared using a one-way ANOVA with post-hoc Bonferroni correction. *p≤0.01.

## ATP-linked respiration rate

ATP-linked respiration rates of PBMCs were calculated in cells taken from healthy controls and moderately affected and severely affected ME/CFS patients. Four aspects of ATP-linked respiration rates from frozen PBMCs incubated in low (1mM) and high (10mM) extracellular glucose were compared–total ATP-linked respiration rate($ATP_{tot}$); glycolysis ATP-linked respiration rate($ATP_{glyc}$); OXPHOS ATP-linked respiration rate ($ATP_{ox}$); % contribution of glycolysis to overall ATP-linked respiration rate ([Fig 5]). Comparisons between the three cohorts showed the severely affected cohort to be significantly lower than the healthy controls in all four parameters in both low and high glucose ($p \leq 0.024$). The moderately affected ME/CFS cohort was shown to be significantly lower than healthy controls in three of the four parameters ($ATP_{glyc}$; $ATP_{ox}$; $ATP_{tot}$) in both low and high glucose conditions ($p \leq 0.007$) but the groups were not different in terms of % contribution of glycolysis to overall ATP-linked respiration rate ($p = 0.271$). In low glucose conditions, when the moderately and severely affected disease cohorts were compared there were no significant differences between the groups ($p \geq 0.126$). However, in high glucose differences between the two disease cohorts were observed with the severely affected patients having significantly lower $ATP_{tot}$ ($p < 0.001$), $ATP_{glyc}$ ($p = 0.002$), and % contribution of glycolysis to overall ATP-linked respiration rate ($p = 0.007$). When looking at the effect of glucose concentration on ATP-linked respiration rate parameters in the three cohorts, significant differences were observed in the healthy control and moderate ME/CFS cohorts. In the healthy control cohort incubating cells in low glucose caused decreases in $ATP_{tot}$, $ATP_{glyc}$, and % contribution of glycolysis to overall ATP-linked respiration rate when compared with cells incubated in high glucose ($p \leq 0.018$). $ATP_{ox}$ increased in healthy controls cells incubated in low glucose ($p < 0.001$). For the moderately affected ME/CFS cohort, both $ATP_{glyc}$ and % contribution of glycolysis to overall ATP-linked respiration rate were shown to be significantly reduced in low glucose conditions ($p \leq 0.049$). The severely affected cohort showed no significant differences between ATP-linked respiration rates of cells incubated in low and high extracellular glucose ($p \geq 0.744$).

## Discussion

In this study, we analysed data previously published by our group [6] in order to see how disease severity in ME/CFS relates to mitochondrial and glycolytic function in PBMCs. We have also calculated ATP-linked respiration rate and glycolysis using new techniques. Previous studies have shown there to be a correlation between blood cell bioenergetic capacity and both aerobic and anaerobic activity [10, 23].

There were no differences between the moderately and severely affected ME/CFS groups for any of the cellular respiratory parameters ([Fig 1]). This suggests that mitochondrial function of PBMCs does not correlate with disease severity in ME/CFS and even those moderately affected by the disease have an impairment in bioenergetic function. The lack of association between disease severity and mitochondrial function suggests that the previously reported reduction in mitochondrial function in ME/CFS is not due to deconditioning. If the lower mitochondrial functioning reported in ME/CFS was due to deconditioning we would expect to see a significantly lower mitochondrial function in the severely affected cohort who were housebound or bedbound when compared to the moderately affected cohort [24, 25].

Nor is it due to the presence of deleterious mtDNA variants. A prior paper did not find mtDNA differences between moderately and severely affected patients despite showing differences between these two groups and controls [26] Thus this work provides important experimental evidence to support one of the few genetic studies conducted on patients with ME/CFS.

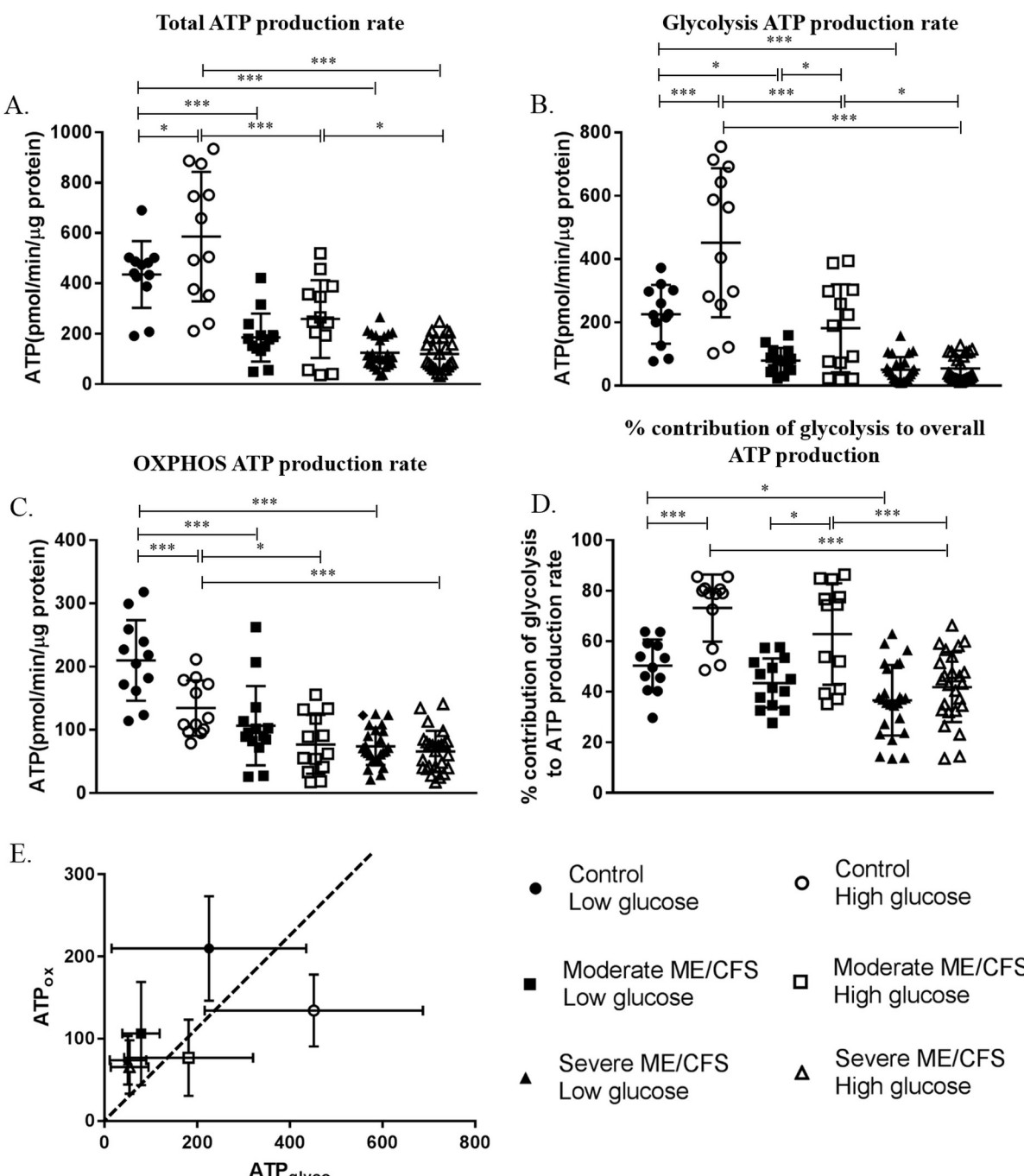

**Fig 5. ATP-linked respiration rates in PBMCs from healthy controls and moderately and severely affected ME/CFS patients in low (1mM) and high (10mM) glucose conditions. (A)** Total ATP-linked respiration rate. **(B)** Glycolysis ATP-linked respiration rate. **(C)** OXPHOS ATP-linked respiration rate. **(D)** % contribution of glycolysis to overall ATP-linked respiration rate. **(E)** Each group plotted as ATP produced via glycolysis ($ATP_{glyco}$) against ATP produced via OXPHOS ($ATP_{OX}$). The dotted line represents equal production of ATP from glycolysis and OXPHOS. Groups to the right of the line are primarily glycolytic (>50% of ATP derived from glycolysis pathway) while groups to the left of the line use OXPHOS as the primary pathway for ATP-linked respiration (>50% of ATP produced via OXPHOS). Groups were compared using two-way ANOVAs with post-hoc Bonferroni correction. Healthy controls low glucose n = 12; Healthy controls high glucose n = 12; moderately affected ME/CFS low glucose n = 14; moderately affected ME/CFS high glucose n = 13; severely affected ME/CFS low glucose n = 25; severely affected ME/CFS low glucose n = 25. $^*$p≤0.01; $^{***}$p≤0.001.

We examined the relative contributions of glycolytic and respiratory acidification to overall extracellular acidification in PBMCs from ME/CFS patients and healthy controls. Glycolytic acidification was shown to be significantly lower in the ME/CFS cohort (Fig 3). This suggests that the contribution of respiratory acidification in ME/CFS PBMCs is significantly higher than in healthy control cells. However, when the ME/CFS cohort was stratified into moderately and severely affected patients, we showed that the severely affected patient group were the ones with significantly lower glycolytic acidification, while the moderately affected cohort had levels of glycolytic acidification comparable to those of healthy controls. This shows this importance of accurately determining glycolytic acidification before calculating glycolytic parameters. We have shown that when calculating glycolysis as directed by the manufacturer's there are no significant differences between the controls, moderately affected patients, and severely affected patients. However, when taking into account respiratory acidification we see that glycolysis is significantly lower in the severely ME/CFS cohort (Fig 4). The data presented here is in agreement with a metabolic profiling study which showed a reduction in specific metabolic markers in blood and urine of ME/CFS patients, pointing towards a reduction in glycolytic activity in ME/CFS [27]. However, caution must be used when interpreting these results as the lower glycolytic function may be representative of deconditioning in the severely affected group. In order to confirm or refute this, further studies are needed.

When we examined the ATP-linked respiration rates of PBMCs from ME/CFS patients and healthy controls, and the effect of glucose concentration on the ATP-linked respiration rates, we found significant differences between the healthy control and ME/CFS cohorts (Fig 5). Lower ATP-linked respiration rates were seen in both the moderately and severely affected ME/CFS cohorts compared to the healthy controls. This reflects the previously published work, which used calculations from a mitochondrial stress test to approximate ATP-linked respiration [6]. These results indicate that there is a defect in the glycolysis and/or OXPHOS pathways that leaves patients cells unable to produce adequate levels of ATP. This may contribute to the symptoms of fatigue and post-exertional malaise which are the hallmarks of ME/CFS. The severely affected cohort had significantly lower $ATP_{glyc}$ and $ATP_{tot}$ than the moderately affected cohort in high glucose. The differences in the rates of ATP-linked respiration between the two patient cohorts may explain why the severely affected patients are rendered bedbound/ housebound by the disease as they have even lower ATP-linked respiration rates than those moderately affected by the disease. However, whether the differences are a cause or consequence of the disease remains to be determined.

We have shown that differences in glucose concentration affected ATP production rates from glycolysis and OXPHOS in healthy controls cells. The changes in control PBMCs with low glucose concentrations followed the pattern we hypothesised–a decrease in glycolysis and an increase in OXPHOS utilisation. This is because the lower glucose concentration pushes cells to use OXPHOS to a greater extent due to the higher yield of ATP per glucose molecule compared with glycolysis. The moderately affected patient cohort shows the same trends for all four parameters as the controls, although not all differences were significant. The severely affected cohort showed very little difference between the low and high glucose groups. We have shown that the control and moderately affected PBMCs are glycolytic in high glucose conditions but shift towards OXPHOS as the predominant source of energy in low glucose conditions. Cells from severely affected patients produce their ATP primarily from cellular respiration (OXPHOS) in both low and high glucose conditions with very little change in energy production between the two glucose conditions. It may be that ME/CFS patients, particularly the severely affected cohort, are already producing ATP at their maximum rate in the high glucose conditions, therefore lowering the glucose concentration has no effect of ATP-linked respiration rates. This may indicate abnormalities in factors such as glucose transport, glycolytic/

OXPHOS enzyme activity, or AMPK function which has previously been shown to be abnormal in ME/CFS skeletal muscle cells [28]. It may also be the case that ME/CFS patient PBMCs are less able to adapt or are slower at adapting to the low glucose environment.

There are several limitations of this study. PBMCs are a very heterogeneous cell population that are comprised of a number of different cell types with different bioenergetic demands [29]. In this study we have used PBMCs rather than stratifying the population out into the subpopulations of cells that PBMCs are comprised of. This was due to the high number of cells required to look at each of the PBMC sub-populations in turn. However, if ethical approval could be obtained in order to collect enough cells to do so, then this stratification would add strength to the study. Future studies should also look at lipid metabolism (which has been shown in other studies to be altered in ME/CFS [30–32]) and whether it differs between the cohorts and how it compares to results seen here in glucose metabolism. The limitation of just investigating glucose metabolism in these cells means that an overall assessment of cellular bioenergetics cannot be made. Valuable information would be gained from repeating these types of experiments and analyses in other cells types in order to see whether these changes are specific to PBMCs or are more systemic. This work would also benefit from being conducted in other disease cohorts with fatigue as a core symptom in order to determine whether the changes in bioenergetics are a marker of the disease ME/CFS or the symptom of fatigue. While clear differences between cohorts have been identified in this study, even when using robust and conservative statistical techniques, higher participant numbers, including samples from multiple international locations, would strengthen the findings.

## Conclusions

It is vital that people with different disease severities are included in ME/CFS studies if we are to move forward with better understanding of ME/CFS pathophysiology. The inclusion of the two disease severity patient groups has allowed us to identify differences and similarities between those moderately and severely affected with the disease. The lack of association between disease severity and mitochondrial function shown here indicates that abnormalities in mitochondrial function are a feature of the disease irrespective of severity. The lower glycolytic functioning in the severely affected patient group that we have identified is also vital as it shows that these patients have a glycolytic impairment in addition to the mitochondrial impairment which may explain why these patients present with a more severe phenotype. Lower levels of both mitochondrial and glycolytic functioning may be caused by a hypometabolic state in ME/CFS which is linked to disease severity. This work has increased our understanding of cellular energy production abnormalities in ME/CFS and how this alters with disease severity.

## Supporting information

**S1 Appendix. Correlation graphs for mitochondrial stress test parameters calculated in PBMCs from healthy controls, moderately affected ME/CFS patients, and severely affected ME/CFS patients.**
(DOCX)

**S2 Appendix. Glycolytic parameters adjusted for respiratory acidification.** These graphs show the rates of glycolysis, glycolytic capacity, glycolytic reserve, and non-glycolytic acidification when respiratory acidification had been taken into account.
(DOCX)

## Acknowledgments

Thanks go to Donna McAvoy for collecting blood samples used in this study, Loranne Agius for her expert advice, and the FCCF at Newcastle University for providing access to the extra-cellular flux analyser. We would like to thank all of the participants who contributed to the original study.

## Author Contributions

**Conceptualization:** Cara Tomas.

**Data curation:** Cara Tomas.

**Formal analysis:** Cara Tomas, Joanna L. Elson.

**Funding acquisition:** Julia L. Newton, Mark Walker.

**Investigation:** Cara Tomas.

**Methodology:** Cara Tomas.

**Resources:** Victoria Strassheim, Julia L. Newton, Mark Walker.

**Supervision:** Joanna L. Elson, Julia L. Newton, Mark Walker.

**Writing – original draft:** Cara Tomas.

**Writing – review & editing:** Cara Tomas, Joanna L. Elson, Victoria Strassheim, Julia L. Newton, Mark Walker.

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
