## [Decision Letter · Decision Letter 0]

28 Jan 2020

PONE-D-19-34254

The effect of Myaglic Encephalomyelitis/chronic fatigue syndrome (ME/CFS) severity on cellular bioenergetic function

PLOS ONE

Dear Dr. Tomas:

Thank you for submitting your manuscript to PLOS ONE. After careful consideration, we feel that it has merit but does not fully meet PLOS ONE’s publication criteria as it currently stands. Therefore, we invite you to submit a revised version of the manuscript that addresses the points raised during the review process.

We would appreciate receiving your revised manuscript by July 22, 2020. To enhance the reproducibility of your results, we recommend that if applicable you deposit your laboratory protocols in protocols.io, where a protocol can be assigned its own identifier (DOI) such that it can be cited independently in the future. For instructions see: http://journals.plos.org/plosone/s/submission-guidelines#loc-laboratory-protocols

We look forward to receiving your revised manuscript.

Kind regards,

Jianhua Zhang

Academic Editor

PLOS ONE

Reviewers' comments:

Reviewer's Responses to Questions

**Comments to the Author**

1. Is the manuscript technically sound, and do the data support the conclusions?

Reviewer #1: No

Reviewer #2: Yes

Reviewer #3: Partly

2. Has the statistical analysis been performed appropriately and rigorously? 

Reviewer #1: Yes

Reviewer #2: Yes

Reviewer #3: No

3. Have the authors made all data underlying the findings in their manuscript fully available?

Reviewer #1: Yes

Reviewer #2: Yes

Reviewer #3: No

4. Is the manuscript presented in an intelligible fashion and written in standard English?

Reviewer #1: Yes

Reviewer #2: Yes

Reviewer #3: No

5. Review Comments to the Author

Reviewer #1: The manuscript “The effect of Myalgic Encephalomyelitis/chronic fatigue syndrome (ME/CFS) severity on cellular bioenergetic function” by Tomas et al. re-analyzed previously published data to determine the impact of severity on glycolytic and mitochondrial function in peripheral blood mononuclear cells (PBMCs). The authors divide up their previous ME/CFS cohort into “moderate” and “severe” and utilize new calculations to determine ATP production rates and distinguish between glycolytic acidification and respiratory acidification. While the concept of teasing out the relevance of disease severity of PBMC bioenergetic function, this study does not advance enough upon the previous publication to justify being a stand-alone manuscript. Major concerns are listed out below.

1) As the authors state, the heterogeneity of this disease is wide-ranging. As such, the line between moderate and severe forms of this disease would seem to need to be better defined aside from just the ability to come into the fatigue clinic. Perhaps a more thorough description in the methods/results as to the criteria chosen (including those outlined in the NICE guidelines), and how these criteria might influence how and when PBMCs were isolated (i.e. age, sex, stage of disease) would all provide critical information that is missing.

2) Claiming that the severe group significantly delineates from the moderate group (i.e. in Figure 2B) seems to also be attributable to a few of the more severe cases, as ~40% of individuals still cluster in a similar range to the moderate cohort. Along these lines, there also is some bias towards to the severe cohort, as 25 patient PBMCs fall into the severe group, whereas only 13 fall into the moderate group. As such, it is hard to fully determine which of the 25 individuals is the best representation of this cohort.

3) The difference between glycolytic acidification and respiratory acidification, and how they are determined, is also not made very clear. A better description of how the authors came to differentiate between the two is needed.

Reviewer #2: The study by Tomas et al, investigated whether Myalgic Encephalomyolitis/chronic Fatigue syndrome (ME/CFS) severity correlates with mitochondrial and glycolytic functions. The authors re-examined previously published bioenergetics data on peripheral blood mononuclear cells (PBMCs) using an equation described by another research group, which accounts for both the respiratory acidification and the glycolytic acidification when calculating glycolytic parameters from a glycolysis stress test. The authors using these equations investigated total ATP production rate, the percentage contribution of glycolysis and OXPHOS to this total ATP along with glycolysis rates and mitochondrial oxygen consumption parameters. The authors show primarily that both ME/CFS cohorts, moderately and severely affected patients, show evidence of mitochondrial dysfunction, thus concluding that disease severity does not correlate with mitochondrial function. The authors however discriminate the two cohorts based on glucose metabolism, illustrating glycolytic impairments only in the severely affected cohort. While the findings are interesting in highlighting distinct bioenergetics profiles within ME/CFS populations, the following points should be addressed:

Major points:

1) The innovation is not clear since prior studies have underlined both glycolytic and mitochondrial dysfunctions as contributors to the illness. How is this study contributing in a better understanding of ME/CFS physiopathology? The citations undervalue prior works related to the approach in general (eg: Shiva, Darley-Usmar, Molina groups among others)

2) The authors have highlighted in the discussion the importance of repeating these experiments on other cell types. My question is along those lines, given that PBMCs are metabolically quiescent cells; how are these findings relevant to the disease mechanisms? Are the authors confident that their findings would still be pertinent in metabolically active cells?

3) The authors have focused their study primarily on mitochondria respiration rates and glyscolysis; and have switched the cells to low glucose to assess the effect of glucose concentration rates on ATP production. Have the authors considered studying the lipid metabolism? Are mitochondria reliant on fatty acid beta-oxidation in this model? If so, is this response also different between the two ME/CFS Cohorts? This limitation should be addressed or at least discussed.

4) The authors suggest that low glycolysis rates may explain the severity of the disease in the severely affected group. If that is the case, the study and its conclusion would benefit from modeling altered glycolysis function (pdh mutations, PDH silencing) in moderately affected cells to see if it mimics disease severity.

Reviewer #3: This interesting article reanalyzes previously published data of PBMC bioenergetics in fatigue syndrome. As technologies and ideas change this is a reasonable approach. The different analysis does appear to reveal some new aspects so this is a worthwhile contribution to the literature. However, it needs to go a bit further with a more comprehensive approach including new information that the bioenergetic profiles revealed by these analyses in human populations is an integrated program. Once these analyses are included a clearer picture may emerge. The impact will be improved by a more comprehensive overview and context which at the moment is minimal.

Major points

1) A comprehensive multi variate analysis is important since it may reveal changes in the overall bioenergetic program as shown in Mitochondria in precision medicine; linking bioenergetics and metabolomics in platelets. Redox Biol 22, 101165 2019 An advantage of this approach is that you do not need to separate into moderate or severe which increases statistical power-which maybe the real reason you do not see a relationship to disease severity. You can also include a clinical parameter in the analysis to test this.

2) A BHI calculation may also be revealing here.

3) Do not understand Figure 3 -The with and without glucose experiment is confusing. Some clearer description of the original experiment is needed so readers do not need to read both. Some justification of using frozen cells is needed-the mitochondria cannot make ATP after freeze thawing so is the valid?

Minor Points

1)PBMC are a heterogenous mixture of cells with different bioenergetic programs (see Redox Biology 2:206-210 2014). This limitation needs to be noted.

2) The citations are woefully inadequate and do not reflect the current thinking in the field of translational bioenergetics or the significant contributions from other groups. Please extend the introduction and discussion significantly to address this.

3) Materials and Methods need to be more complete defining the method for PBMC isolation and characterization even though this maybe in the previous publication.

4) Strictly speaking this is cellular respiration not mitochondrial since these are not isolated mitochondria and other factors contribute to the control of respiration which are not mitochondrial-please change this terminology. (see ref in comment 2). This is not ATP production but ATP linked respiration (see Biological chemistry 401 (1), 3-29 2019).

5) A section is needed on normalization and the definition of parameters-eg coupling efficiency.

6) The statistical markings are missing from 2A. Does not look significant but text says it is.

7) No need for sentence 191 -this is not the guideline but the published papers have for some time shown this understanding of the different processes contributing to ECAR

8) The conclusions are not really conclusions but a list of what was done-rewrite to discuss conclusions

6. PLOS authors have the option to publish the peer review history of their article (what does this mean?). If published, this will include your full peer review and any attached files.

Reviewer #1: No

Reviewer #2: No

Reviewer #3: No

---

## [Author Response · Author response to Decision Letter 0]

22 Feb 2020

We would like to sincerely thank all three of the reviewers for their comments. We have addressed both the major and minor comments raised by the reviewers. Substantial changes have been made including additions to the introduction to better reflect the literature surrounding the techniques utilised in the manuscript. The methods section has been altered to better describe how cells were isolated, the experimental technique of extracellular flux analysis, and to describe more clearly the accepted parameters for determining disease severity in ME/CFS patients. This has been an area of intense debate. A new figure and table have been added to the manuscript to reflect analysis that has been completed following a suggestion from reviewer 3. This added analysis looks at the relationships between mitochondrial stress test parameters in the three cohorts. The discussion and conclusions have also been amended after considering comments from the reviewers.

Changes are itemised below it italics and are highlighted in the track changes version of the manuscript.

Reviewer #1: 

1) As the authors state, the heterogeneity of this disease is wide-ranging. As such, the line between moderate and severe forms of this disease would seem to need to be better defined aside from just the ability to come into the fatigue clinic. Perhaps a more thorough description in the methods/results as to the criteria chosen (including those outlined in the NICE guidelines), and how these criteria might influence how and when PBMCs were isolated (i.e. age, sex, stage of disease) would all provide critical information that is missing.

Author’s comment: We thank the reviewer for their comment. We have now amended the methods section to make the differences between the disease severity groups clearer. 

2) Claiming that the severe group significantly delineates from the moderate group (i.e. in Figure 2B) seems to also be attributable to a few of the more severe cases, as ~40% of individuals still cluster in a similar range to the moderate cohort. Along these lines, there also is some bias towards to the severe cohort, as 25 patient PBMCs fall into the severe group, whereas only 13 fall into the moderate group. As such, it is hard to fully determine which of the 25 individuals is the best representation of this cohort.

Author’s comment: We have now revised to text to more accurately reflect the data by stating that it is only a sub-section of the severely affected cohort that shows the decrease in glycolytic acidification compared to the moderately affected cohort. 

3) The difference between glycolytic acidification and respiratory acidification, and how they are determined, is also not made very clear. A better description of how the authors came to differentiate between the two is needed.

Author’s comment: We have added a section to the text to clarify the calculation of respiratory and glycolytic acidification.

Reviewer #2: 

Major points:

1) The innovation is not clear since prior studies have underlined both glycolytic and mitochondrial dysfunctions as contributors to the illness. How is this study contributing in a better understanding of ME/CFS physiopathology? The citations undervalue prior works related to the approach in general (eg: Shiva, Darley-Usmar, Molina groups among others).

Author’s comment: We have added sections to the introduction and conclusion to outline the importance of this work, particularly the inclusion of severely affected patients in the study. We have also updated the introduction and discussion to better reflect prior work in this area. 

2) The authors have highlighted in the discussion the importance of repeating these experiments on other cell types. My question is along those lines, given that PBMCs are metabolically quiescent cells; how are these findings relevant to the disease mechanisms? Are the authors confident that their findings would still be pertinent in metabolically active cells?

Author’s comment: We thank the reviewer for their comment. As we showed in our previous publication with this data-set, PBMCs from ME/CFS patients are shown to have significantly different bioenergetic profiles than those of healthy controls. While PBMCs may not be the most metabolically active cells, the fact that we have shown there to be differences between the way in which ME/CFS PBMCs and healthy control PBMCs utilise glucose shows their importance. This preliminary work is vital for building an overall picture of fuel utilisation in ME/CFS, which will ultimately include different cell types. PBMCs are more easily accessible than most other cell types, which is why the initial study utilised their availability. Moving forward, we will look at fuel utilisation of other cell types in order to see if the differences seen in PBMCs can also be identified in other tissues. 

3) The authors have focused their study primarily on mitochondria respiration rates and glyscolysis; and have switched the cells to low glucose to assess the effect of glucose concentration rates on ATP production. Have the authors considered studying the lipid metabolism? Are mitochondria reliant on fatty acid beta-oxidation in this model? If so, is this response also different between the two ME/CFS Cohorts? This limitation should be addressed or at least discussed.

Author’s comment: As this was a re-analysis of the previously published data that did not look at lipid metabolism, therefore lipid metabolism was not included in this manuscript. We are aware of the importance of lipid metabolism in ME/CFS that has been reported in a number of manuscripts looking at metabolism in those with ME/CFS. Other studies we have conducted have looked at this aspect of ME/CFS cellular bioenergetic function but were not included in the scope of this manuscript. We have added a section to the discussion to address this limitation. 

4) The authors suggest that low glycolysis rates may explain the severity of the disease in the severely affected group. If that is the case, the study and its conclusion would benefit from modeling altered glycolysis function (pdh mutations, PDH silencing) in moderately affected cells to see if it mimics disease severity.

Author’s comment: As discussed previously, the purpose of this study was to assess the previously published data using new techniques. This manuscript describes exciting new findings including the ability of severely affected patients to utilise the glycolysis pathway and mitochondria for ATP production. 

Reviewer #3: 

Major points

1) A comprehensive multi variate analysis is important since it may reveal changes in the overall bioenergetic program as shown in Mitochondria in precision medicine; linking bioenergetics and metabolomics in platelets. Redox Biol 22, 101165 2019 An advantage of this approach is that you do not need to separate into moderate or severe which increases statistical power-which maybe the real reason you do not see a relationship to disease severity. You can also include a clinical parameter in the analysis to test this.

Author’s comment: We have looked at the reference you recommended and completed the types of analysis described in the paper. We have added another figure and table to the manuscript to show the newly analysed data (figure 2). The new figure shows the most significant correlations with the graphs for all other correlations can be seen in the supplementary data file. We have also added a section to the paper to describe the newly conducted analysis.

2) A BHI calculation may also be revealing here.

Author’s comment: We thank the reviewer for their suggestion and this is something we considered including, however, 13 of our samples have a negative reserve capacity, which means that a BHI cannot be calculated. We were reluctant to embark on new analysis that would require discarding this number of samples.

3) Do not understand Figure 3 -The with and without glucose experiment is confusing. Some clearer description of the original experiment is needed so readers do not need to read both. Some justification of using frozen cells is needed-the mitochondria cannot make ATP after freeze thawing so is the valid?

Author’s comment: We thank the reviewer for their comment regarding the lack of clarity surrounding figure 3. The figure shows the glycolysis of cells as reported in the previous manuscript (Fig 3A) and after that, data has been adjusted for glycolytic acidification. We have amended the text to make this clearer and given a brief description of how glycolysis was measured in the previous publication. 

Our previous publication investigated the effect of freezing on PBMC mitochondrial function and found “the absolute values of some of the parameters are affected by the freezing process; [however] the pattern can be seen in both cohorts and therefore either fresh or frozen samples can be used to detect differences between control and CFS cohorts.” We also concluded, “frozen samples are adequate to show the differences between the cohorts sufficiently”. Given that the effect of freezing is something that we considered very carefully in our original study, we are confident that our results from frozen cells are valid. 

Minor Points

1)PBMC are a heterogenous mixture of cells with different bioenergetic programs (see Redox Biology 2:206-210 2014). This limitation needs to be noted.

Author’s comment: We have added a comment addressing this in the discussion. 

2) The citations are woefully inadequate and do not reflect the current thinking in the field of translational bioenergetics or the significant contributions from other groups. Please extend the introduction and discussion significantly to address this.

Author’s comment: We apologise for the absence of key work within the field. We have updated the text to better reflect the key prior work. 

3) Materials and Methods need to be more complete defining the method for PBMC isolation and characterization even though this maybe in the previous publication.

Author’s comment: We have now given a more thorough description of this process in the methods section.

4) Strictly speaking this is cellular respiration not mitochondrial since these are not isolated mitochondria and other factors contribute to the control of respiration which are not mitochondrial-please change this terminology. (see ref in comment 2). This is not ATP production but ATP linked respiration (see Biological chemistry 401 (1), 3-29 2019).

Author’s comment: We have amended the text to use the terminology suggested. 

5) A section is needed on normalization and the definition of parameters-eg coupling efficiency.

Author’s comment: This has been added to the methods section.

6) The statistical markings are missing from 2A. Does not look significant but text says it is.

Author’s comment: We included the statistical markings on the graph for Fig 2A so we are unsure as to why they did not appear on the version seen by reviewer 3. 

7) No need for sentence 191 -this is not the guideline but the published papers have for some time shown this understanding of the different processes contributing to ECAR

Author’s comment: This sentence has been removed. 

8) The conclusions are not really conclusions but a list of what was done-rewrite to discuss conclusions

Author’s comment: The conclusions section has been re-written in order to be more focused on the conclusions that can be drawn from this work.

---

## [Decision Letter · Decision Letter 1]

18 Mar 2020

The effect of Myalgic Encephalomyelitis/chronic fatigue syndrome (ME/CFS) severity on cellular bioenergetic function

PONE-D-19-34254R1

Dear Dr. Tomas

We are pleased to inform you that your manuscript has been judged scientifically suitable for publication and will be formally accepted for publication once it complies with all outstanding technical requirements.

With kind regards,

Jianhua Zhang

Academic Editor

PLOS ONE

Additional Editor Comments (optional):

Reviewers' comments:

Reviewer's Responses to Questions

**Comments to the Author**

1. If the authors have adequately addressed your comments raised in a previous round of review and you feel that this manuscript is now acceptable for publication, you may indicate that here to bypass the “Comments to the Author” section, enter your conflict of interest statement in the “Confidential to Editor” section, and submit your "Accept" recommendation.

Reviewer #1: All comments have been addressed

Reviewer #2: All comments have been addressed

Reviewer #3: All comments have been addressed

2. Is the manuscript technically sound, and do the data support the conclusions?

Reviewer #1: (No Response)

Reviewer #2: Yes

Reviewer #3: Yes

3. Has the statistical analysis been performed appropriately and rigorously? 

Reviewer #1: (No Response)

Reviewer #2: Yes

Reviewer #3: Yes

4. Have the authors made all data underlying the findings in their manuscript fully available?

Reviewer #1: (No Response)

Reviewer #2: Yes

Reviewer #3: Yes

5. Is the manuscript presented in an intelligible fashion and written in standard English?

Reviewer #1: (No Response)

Reviewer #2: Yes

Reviewer #3: Yes

6. Review Comments to the Author

Reviewer #1: (No Response)

Reviewer #2: The authors have made the recommended changes to the manuscript

The paper is acceptable for publication

No further comments

Reviewer #3: The revision is excellent and the presentation of the data in Figure 2 intriguing! The paper will be an important contribution to the field.

7. PLOS authors have the option to publish the peer review history of their article (what does this mean?). If published, this will include your full peer review and any attached files.

Reviewer #1: No

Reviewer #2: No

Reviewer #3: No

---

## [Editor Report · Acceptance letter]

25 Mar 2020

PONE-D-19-34254R1 

The effect of Myalgic Encephalomyelitis/chronic fatigue syndrome (ME/CFS) severity on cellular bioenergetic function 

Dear Dr. Tomas:

I am pleased to inform you that your manuscript has been deemed suitable for publication in PLOS ONE. Congratulations! Your manuscript is now with our production department. 

With kind regards,

on behalf of

Dr Jianhua Zhang 

Academic Editor

PLOS ONE